# Numerical Investigation of the Influence of Fatigue Testing Frequency on the Fracture and Crack Propagation Rate of Additive-Manufactured AlSi10Mg and Ti-6Al-4V Alloys

**Mustafa Awd *** and **Frank Walther**

Chair of Materials Test Engineering (WPT), TU Dortmund University, Baroper Str. 303,
44227 Dortmund, Germany; frank.walther@tu-dortmund.de
*   Correspondence: mustafa.awd@tu-dortmund.de

**Abstract:** Advances in machine systems and scanning technologies have increased the use of selective laser melted materials in industrial applications, resulting in almost full-density products. Inconsistent mechanical behavior of components under cyclic stress is caused by microstructure and porosity created during powder melting. The extended finite element method, XFEM, was used to imitate crack propagation utilizing an arbitrary fracture route to study fatigue crack growth in additively produced fatigue specimens. The influence of loading level and testing frequency on fatigue life was studied using fracture energy rate curves. Micro-computed tomography (μ-CT) scans offer 2D images in angular increments. There are several ways to reduce the number of faces and vertices. Opensource software was used to isolate the cylindrical shell from interior pores and create finite element models from μ-CT projections. All simulations were on supposedly cylindrical fatigue specimens made by selective laser melting (SLM) based on previous experimental results of the authors. Crack propagation rate curves were utilized to evaluate the effects of loading level and testing frequency. At larger loads, the fracture area increases abruptly at 3E3 cycles, then stabilizes at 4E4 cycles in Al alloys in comparison to Ti-6Al-4V alloys. Crack propagation rate curves may be used to determine Paris constants based on the applied stresses.

**Keywords:** selective laser melting; microstructure; porosity; very high-cycle fatigue; crack propagation; extended finite element method

## 1. Introduction

Fatigue damage and fracture propagation rate are both influenced by the frequency of fatigue tests. Environment and load frequency (0.4 to 58 Hz) impacted the macro fatigue fracture propagation law in 7075-T6 and 2024-T3 Alclad aluminum. The rate of crack propagation reduced as the frequency of loading increased [1]. A range of loading frequencies, from 0.017 Hz up to 140 Hz, was used to corroborate this for both 2024-T3 aluminum and SM-50 steel. Furthermore, the space between striations decreased as the frequency of testing increased [2]. For Ti-8Al-1Mo-1V, a similar result was seen at frequencies between 0.5 and 30 Hz, but it was not ascribed to the frequency of loading but to an impact of ambient hydrogen at the crack tip, the influence of which depends on the frequency [3]. A crack tip subjected to oxide formation from water vapor has a higher propagation rate, according to Zhu et al. [4,5]. The ultrasonic testing equipment used in this study will subject the specimens to almost vacuum conditions by engulfing the specimen surface with high-pressure dry air. This phenomenon was also seen in the high-strength aluminum alloy 7150 and was associated with a decreasing frequency transition from intergranular to transgranular crack propagation, which was reliant on the grain boundary diffusion of atomic hydrogen near the crack tip [6]. The aluminum alloy AlZnMgCu1.5 was also shown to possess this property [7]. In the case of the aluminum alloy E319, this effect was examined more carefully, and the same findings were obtained. Hydrogen reaches

the crack tip, which is connected to ambient water exposure, which in turn is greater at lower testing frequency [4,5]. The process of fracture initiation for E319 has not changed, despite the fact that minor crack propagation has changed [5]. Testing on titanium indicated that fracture propagation rate is frequency-dependent, and the material's viscoplasticity was used to explain this phenomenon [8]. Higher testing frequencies are associated with increased fatigue strength, which is most noticeable at lower stress levels but completely vanishes at stress ratios greater than 0.8 in the Ti-6Al-4V high cycle fatigue regime. Multiple mechanisms, including strain rate effects, dislocation motion, and decreased main slip systems, are thought to have a role [9]. Frequencies up to 20 kHz were used to test Ti-6Al-4V in the gigacycle regime at various heat treatments. It was shown that frequency had little influence on smooth specimens. All specimens have surface fractures, and the higher the frequency of these fractures, the greater the fatigue strength [10].

In this article, the extended finite element method (XFEM) is verified by comparison with experimental data for simulating fatigue crack propagation. Although the influence of frequency is of great importance to the community, a critically low amount of research work has been found discussing the issue numerically for additively manufactured materials. In addition, the numerical analysis of short fatigue crack transition to long fatigue crack is scarcely found in the literature. In contrast to Ti-6Al-4V, AlSi10Mg was proven to have three distinct damage stages. At first, both alloys showed steady damage evolution with modest damage evolution rates. The anticipated number of cycles to failure was found to be quite accurate, and this information will be utilized to investigate the influence of loading frequency on fatigue lifetime and crack propagation curves.

## 2. Materials and Methods

### 2.1. Materials Processing and Characterization

Cylindrical specimens for AlSi10Mg and Ti-6Al-4V were developed utilizing modified SLM 250 HL and 500 HL equipment. The oxidation of the melt tracks was avoided during production using an inert argon gas environment. The process parameters for a single exposure specimen can be found in Table 1. $P$ is the power $vs$ the scanning speed, $D$ is the laser spot size, $h_t$ the hatch distance, and $t$ the layer thickness. High-quality components with a relative density of more than 99.5% for both alloys were produced using an improved scanning approach and settings. The authors investigated with secondary exposure experiments, which led to significant improvements in relative density as well as modification of microstructure and mechanical properties. The corresponding parameters can be found in [11]. On Schenck PC63M (45 kN) and Instron 8872 (10 kN) servohydraulic systems, low-frequency experiments were carried out for Ti-6Al-4V at 5 Hz and AlSi10Mg at 20 Hz. Nikon X TH 160 X-ray microcomputed tomography (μ-CT) was used to scan the specimens in the gauge length volume at 160 kV. The mean pore size for AlSi10Mg is 43.45 μm, and the standard deviation is 24.42 μm. Corresponding values for Ti-6Al-4V were 68.91 μm and 36.26 μm, respectively. The relative densities for AlSi10Mg and Ti-6Al-4V were 99.94% and 99.97%, respectively, in a specimen with a gauge section diameter of 5 mm. To manage the high-strain rate cyclic deformation temperature caused by deformation, high-frequency tests were performed on a Shimadzu USF-2000A ultrasonic system using pressurized dry air and a pulse:pause ratio of 1:1 [12,13]. More information on specimen shapes and experimental settings are provided in [11,14,15].

The extended finite element methodology (XFEM), which was used to investigate the influence of testing frequency, is a physically linear-elastic method. Therefore, it fundamentally relies on the accurate determination of the elastic modulus of the specific material configuration. This was achieved through the application of micro instrumented indentation and the indirect deduction of the macroscopic elastic modulus to exclude the influence of porosity on the elastic behavior and include only the microstructure. When assessing the microstructural strength of materials, the elastic modulus was an essential mechanical characteristic. The elastic modulus may be calculated using the ultra-micro instrumented indentation test as follows [16],

**Table 1.** Laser scanning parameters used to construct AlSi10Mg and Ti-6Al-4V alloy specimens in a single exposure procedure.

| | Power | Scanning Speed | Spot Size | Hatching Distance | Layer Thickness |
|---|---|---|---|---|---|
| | $P$ (W) | $vs$ (mm/s) | $D$ (mm) | $h_t$ (mm) | $t$ (mm) |
| **AlSi10Mg** | 350 | 1200 | 0.083 | 0.190 | 0.050 |
| **Ti-6Al-4V** | 240 | 1200 | 0.082 | 0.105 | 0.060 |

$$E^{SI} = \frac{\sqrt{\pi}}{2} \frac{S}{\sqrt{A}}, \tag{1}$$

where $A$ is the notch contact area under the maximum load and $S$ is the slope of the unloading curve. $E^{SI}$ is a 'specimen + indenter' modulus in the case of coated surfaces. The elastic modulus $E$ of the sample depends on the elastic modulus of the surface and subsurface ($E_f$ and $E_s$, respectively), and is established using the definition provided below

$$\frac{1}{E^{SI}} = \frac{1}{E^*} + \frac{1}{E_i^*} \tag{2}$$

with

$$E^* = E / \left(1 - v^2\right) \text{ and } E_i^* = E_i / \left(1 - v_i^2\right) \tag{3}$$

where $E^*$ and $E_i^*$ are the elastic moduli of reduction, and $v$ and $v_i$ are the Poisson's ratios of the specimen and the indenter, respectively. Elastic homogenous materials have the same decreased elastic modulus at both the surface and the subsurface, that is $E_f^* = E_s^*$ and $E^*$ represent the value of the sample reduced elastic modulus: $E^* = E_f^* = E_s^*$. The genuine contact area has a direct effect on the hardness and elastic characteristics [17–19].

$$H = \frac{P}{A_c} \tag{4}$$

$$\frac{1 - v^2}{E} = \frac{1}{E_c^*} - \frac{1 - v_i^2}{E_i} \tag{5}$$

$$E_c^* = \frac{S}{2} \sqrt{\frac{\pi}{A_c}}. \tag{6}$$

$H$ is the hardness modulus and $E$ is the elastic modulus of the material. In (4) to (6), $P$ is the applied load, $E_c^*$ is the reduced contact modulus between the tip and the specimen in case of the free surface specimen without coating, and $A_c$ is the projected contact area. In (5) and (6), except for $A_c$, the indentation test can determine or measure all of the characteristics. The predicted contact area of a perfect pyramid penetrator is linked to the contact depth $h_c$ through the geometry of the tip

$$A_c = \pi \cdot tan^2\theta \cdot h_c^2 \tag{7}$$

where $\theta$ is the equivalent half-angle at the vertex of the tip. At depths of fewer than a few hundred nanometers, the blunting effect of the tip will diminish the predicted contact area. To account for tip blunting, Oliver and Pharr created a multi-variable tip area function [20]. Indentation technology cannot identify the eventual accumulation or sinking that occurs throughout the indentation process. Hence, the relationship between the contact depth and the indentation depth h (measured by indentation technology) is not easy to determine. There have been a number of theories proposed to explain this phenomenon [20].

$$h_c = h - \epsilon \frac{P}{S}, \tag{8}$$

where $\epsilon$ = 0.75 is a geometrical factor for a pyramidal indenter, *P* is the maximum load, *S* is the slope of the unloading curve, and *h* is the reading depth on the experiment values. This method does not consider the pile-up simply because in (8), $h_c$ cannot be greater than *h*.

AlSi10Mg and Ti-6Al-4V were tested on a Shimadzu DUH-211 dynamic ultra-micro hardness tester. For AlSi10Mg, a Vickers indenter type was employed with an elastic modulus of $1.14 \times 10^3$ GPa and a Poisson's ratio of 0.07 to demonstrate an example of elastic modulus determination (Figure 1). Equation (7) specifies a fixed 68° semi-apex angle. First, unloading's maximum depth of 1.893 μm was taken into account in the calculation of $h_c$. The elastic modulus for AlSi10Mg in Figure 1a is ~83 GPa since *P* is equal to the highest load after the first unloading. Figure 1b show that the elastic modulus of Ti-6Al-4V is ~112 GPa. The elastic modulus obtained from the conventional tensile test was greater in the case of Ti-6Al-4V but lower in the case of AlSi10Mg [11,21]. The same procedure was used to measure mechanical characteristics for functionally graded binary alloys with second exposure treatments [11]. The benefit of this method was the speed and efficiency with which the effect of process parameters on mechanical qualities could be monitored [22]. The elastoplastic properties of metals and alloys were extracted from the obtained data using an instrumented indentation method in deep neural networks [23]. In order to convert hardness into yield strength, a constraint factor was also utilized [24]. Microstructure flow characteristics may be determined via instrumented indentation, which isolates the effects of defects from process factors.

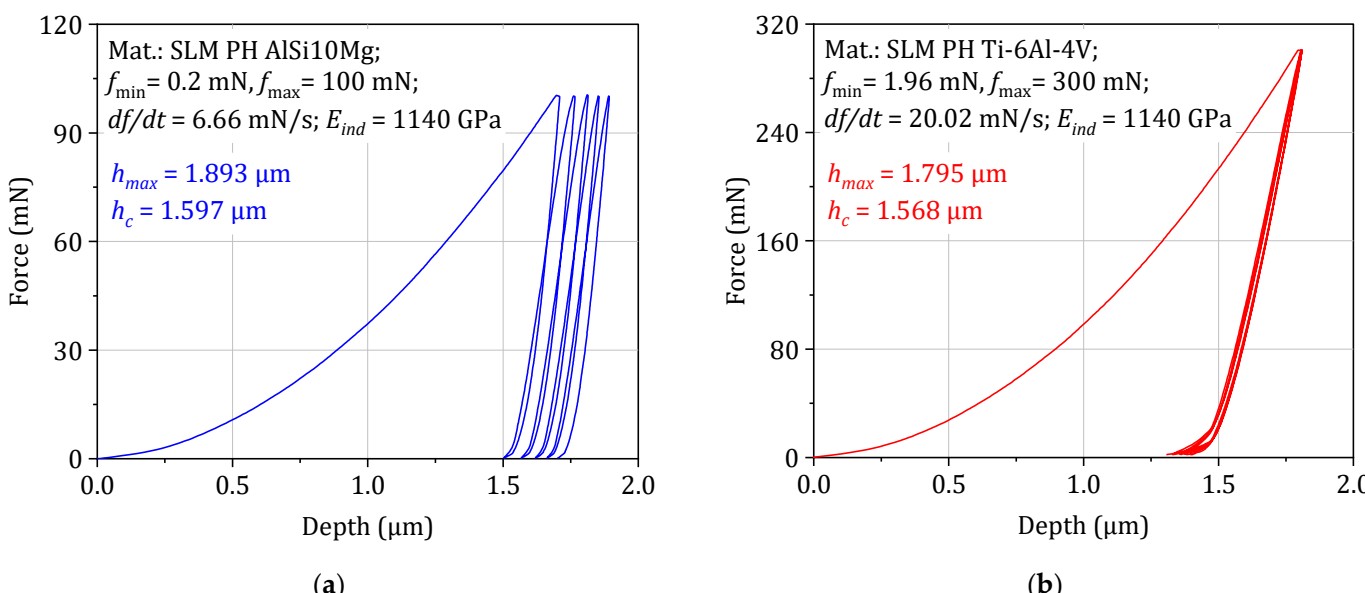

(**a**)  (**b**)

**Figure 1.** Comparison between ultra-micro indentation behavior in (**a**) AlSi10Mg and (**b**) Ti-6Al-4V.

*2.2. The Extended Finite Element Method*

Using the XFEM, we were able to determine the inverse crack propagation rate at low and ultrasonic frequencies. In a linear elastic fracture mechanics (LEFM) framework, only linear elastic characteristics were taken into account. XFEM is able to handle the discontinuous displacement field on the fracture surface and the singularity at the end of the crack. An enrichment function was added to the general finite element technique (GFEM) using the partition of unity method (PUM) [25]. The asymptotic functions at the fracture tip and a discontinuous function to describe the displacement change at the crack surface make up the enrichment function [26]. The displacement vector function of the XFEM may now be represented as a vector

$$\vec{u} = \sum_{I=1}^{N} N_I(x)[\vec{u}_I + H(x)\vec{a}_I + \sum_{\alpha=1}^{4} F_\alpha(x)\vec{b}_I^\alpha] \tag{9}$$

where $H(x)$ is the jump enrichment function following the Heaviside distribution and $\vec{a}_I$ is an enriched nodal degree of freedom (DOF) for jump discontinuity [25]. There is a leap function and a crack tip function for nodes with circles and squares, respectively. The blue element serves as a crack tip, the yellow ones are enhanced ones, and the pink ones are blends of the two, Figure 2.

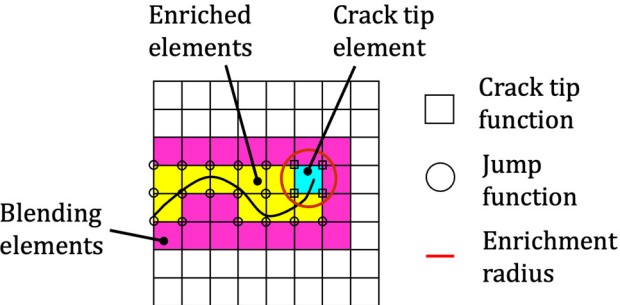

**Figure 2.** XFEM model 2D mesh with enhanced nodes and radius [25].

It is necessary to calibrate the scanner before beginning the process of scanning so that shading inaccuracies may be minimized. VGStudio Max 2.2 Heidelberg, Germany processes raw data from microcomputed tomography (μ-CT) scans to display the 3D fault distribution inside the material. A Python-based CAD application was used to extract and process the specimens' surfaces as well as the inside surfaces of the pores [27]. Numerous simplifications and reduction filters were introduced to the model so that it could be studied using XFEM, Figure 3. As indicated by Awd et al. [11,28], the phase elements of the microstructure of these configurations of AlSi10Mg and Ti-6Al-4V are on the scale of the submicron, which requires a different X-ray setup such as a synchrotron-based X-rays. The study focuses on crack initiation from pores. Therefore, the implemented X-ray setup was considered sufficient.

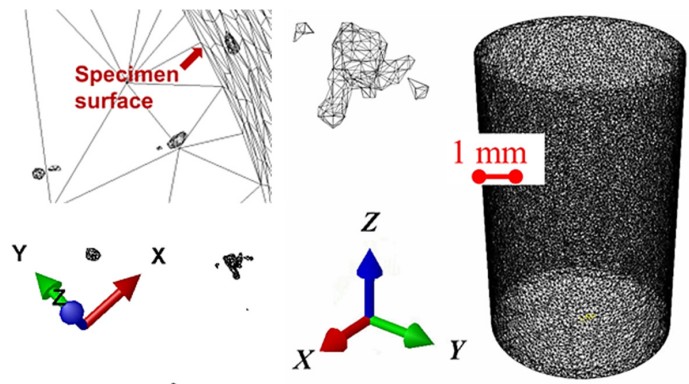

**Figure 3.** STL depiction of a μ-CT model.

As a function of energy, a fracture grows in length when sufficient energy can be provided to overcome the material's resistance. $G$ stands for the amount of energy required to build the fracture [29].

$$G = -\frac{d\Pi}{dA}. \tag{10}$$

$A$ is the area of the crack, and $\Pi$ is the total potential energy [29]. It indicates how fast the energy is released when the crack propagates. The connection between $\Delta G$ and $da/dN$ follows the Paris law [30,31] in the middle region if $\Delta G_{th} < \Delta G < G_{pl}$, as shown by [32]

$$\frac{da}{dN} = c_3(\Delta G)^{c_4} \tag{11}$$

where the parameters $c_3$ and $c_4$, found in Table 2, are fitted from the instrumented indentation tests and experimental fatigue date.

**Table 2.** Energy release rate crack propagation parameters according to Equation (11).

| mm/Cycle | $c_3$ [Cycle$^{-1}\cdot$J$^{-1}$] | $c_4$ [-] |
|---|---|---|
| AlSi10Mg (20 Hz) | $4.2 \times 10^{-5}$ | 1.2 |
| (20 kHz) | $7.6 \times 10^{-7}$ | 1.2 |
| Ti-6Al-4V (5 Hz) | $2.1 \times 10^{-6}$ | 1.685 |
| (20 kHz) | $1.22 \times 10^{-8}$ | 1.685 |

Our finite element model uses tetrahedral elements with four nodes for each element. Figure 4 provide an introduction to computing the coordinates of the intersection point.

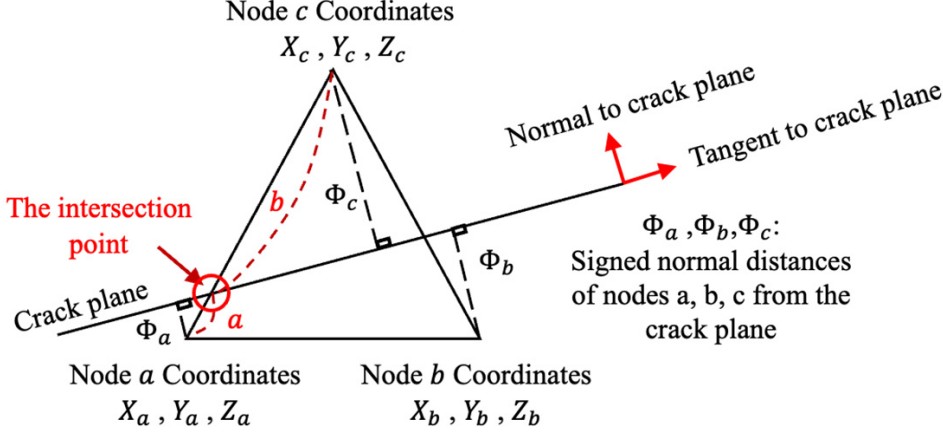

**Figure 4.** A schematic for calculating the junction spots' coordinates.

As shown in Figure 4, the PHILSM values of nodes $a$, $b$, and $c$ are $\Phi_a$, $\Phi_b$, and $\Phi_c$. Therefore, $a/b = |\Phi_a/\Phi_c|$. $a$ and $b$ are the distances between the intersection point and node $a$ and node $c$, respectively. As a result, the intersection point's coordinates may be stated as

$$X = X_a + (X_c - X_a)\cdot|\Phi_a/(\Phi_a - \Phi_c)|$$
$$Y = Y_a + (Y_c - Y_a)\cdot\left|\frac{\Phi_a}{\Phi_a - \Phi_c}\right| \tag{12}$$
$$Z = Z_a + (Z_c - Z_a)\cdot|\Phi_a/(\Phi_a - \Phi_c)|.$$

The intersection point only occurs when the PHILSM values of the two points are in the opposite direction. No matter how positive or negative the two numbers are, they imply that the two points are in agreement. As shown in Figure 5, the shaded region represents a triangle crack section.

By utilizing the points' coordinates (Point 1, Point 2, and Point 3), the triangle's sides $a$, $b$, and $c$ can be determined.

$$a = \sqrt{(X_1 - X_2)^2 + (Y_1 - Y_2)^2 + (Z_1 - Z_2)^2}$$
$$b = \sqrt{(X_2 - X_3)^2 + (Y_2 - Y_3)^2 + (Z_2 - Z_3)^2}$$
$$c = \sqrt{(X_1 - X_3)^2 + (Y_1 - Y_3)^2 + (Z_1 - Z_3)^2} \tag{13}$$
$$s = \frac{a+b+c}{2}$$

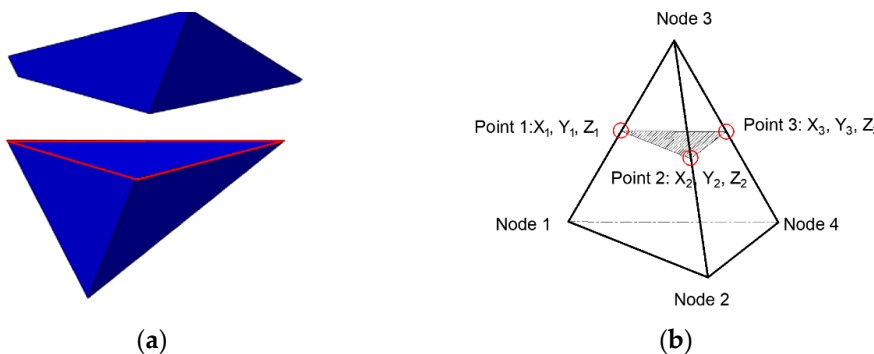

(a)          (b)

**Figure 5.** Crack portion of a triangular crack portion of a triangular crack: (**a**) fractured element; (**b**) intersection points.

Heron's formula may be used to estimate the size of a fracture [33]

$$A = \sqrt{s \cdot (s-a) \cdot (s-b) \cdot (s-c)}. \tag{14}$$

Function myareaTri, written in the Python language, may be used to figure out how big a triangle is using its three vertices' coordinate values as input. Figures 5 and 6 illustrate how to determine the area of a tetrahedral and quadrilateral fracture section (see below).

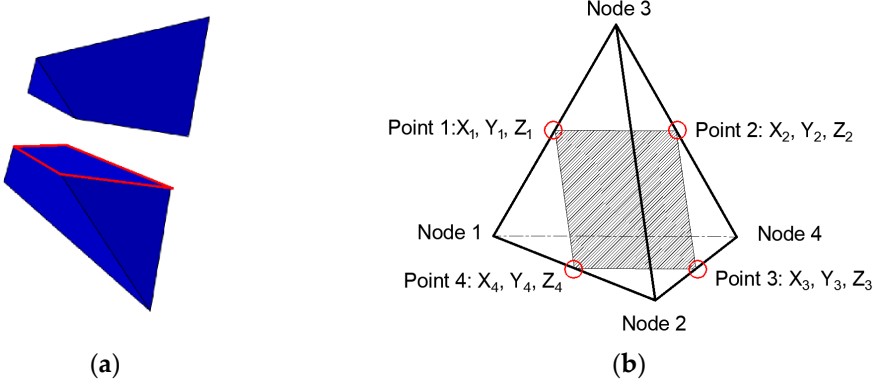

(a)          (b)

**Figure 6.** Crack length and width in a four-cornered crack segment: (**a**) fractured element; (**b**) intersection points.

The sides and diagonals of the quadrilateral crack section are $a, b, c, d, e,$ and $f$.

$$
\begin{aligned}
a &= \sqrt{(X_1 - X_2)^2 + (Y_1 - Y_2)^2 + (Z_1 - Z_2)^2} \\
b &= \sqrt{(X_2 - X_3)^2 + (Y_2 - Y_3)^2 + (Z_2 - Z_3)^2} \\
c &= \sqrt{(X_3 - X_4)^2 + (Y_3 - Y_4)^2 + (Z_3 - Z_4)^2} \\
d &= \sqrt{(X_1 - X_4)^2 + (Y_1 - Y_4)^2 + (Z_1 - Z_4)^2} \\
e &= \sqrt{(X_1 - X_3)^2 + (Y_1 - Y_3)^2 + (Z_1 - Z_3)^2} \\
f &= \sqrt{(X_2 - X_4)^2 + (Y_2 - Y_4)^2 + (Z_2 - Z_4)^2} \\
s_1 &= \frac{a+b+e}{2} \\
s_2 &= \frac{a+d+f}{2} \\
s_3 &= \frac{c+d+e}{2} \\
s_4 &= \frac{b+c+f}{2}
\end{aligned}
\tag{15}
$$

Four triangles were formed, and their areas were calculated using the Python function myareaQuad.

$$
\begin{aligned}
A_1 &= \sqrt{s_1 \cdot (s_1 - a) \cdot (s_1 - b) \cdot (s_1 - e)} \\
A_2 &= \sqrt{s_2 \cdot (s_2 - a) \cdot (s_2 - d) \cdot (s_2 - f)} \\
A_3 &= \sqrt{s_3 \cdot (s_3 - c) \cdot (s_3 - d) \cdot (s_3 - e)} \\
A_4 &= \sqrt{s_4 \cdot (s_4 - b) \cdot (s_4 - c) \cdot (s_4 - f)} \\
A &= (A_1 + A_2 + A_3 + A_4)/2
\end{aligned}
\tag{16}
$$

Vertex coordinates were used as input values for this algorithm. XFEM may be used to model the spread of fatigue cracks in any direction. Based on the virtual crack closure technique (VCCT), the criterion is a fracture-based surface behavior. VCCT in Abaqus [34] is based on the linear elastic fracture mechanics (LEFM) concept, which necessitates a model crack. The given damage initiates criteria control fracture nucleation. The next direct cyclic step may be used to propagate the fracture once it has been nucleated in the static stage. Stress and strain levels must fulfill certain criteria before deterioration may commence. In Abaqus [34], the fatigue crack onset and growth are determined by $\Delta G$ value at the crack tip and $\Delta G$ is calculated by VCCT. For example, in a two-dimensional model, for stabilized cycle $N$, the crack length is $a_N$. $G_{thresh}$ is the threshold energy release rate ($G_{thresh} = 0.01\ G_c$) and $G_{pl}$ is the energy release rate upper limit ($G_{pl} = 0.85\ G_c$). $G_c$ is the J-integral of the indentation which, according to Jeon et al. [35], is equivalent to the plastic work of the indentation. If $G_{max} < G_{thresh}$, the crack will not grow. If $G_{max} > G_{pl}$, the increment no. of cycles ($\Delta N$) is 1. Increasing the cycle number count by 1 will cause the interface elements at the crack tips to be released which means that fracture will occur in the richer elements in front of the crack points. Figure 7 explain data flow, where $f$ is the cut-off tolerance which is carried out at the point of the crack front that leads to the least energy release. $G_{pl}$ is determined from instrumented indentation according to [35]. Additionally, $G_{thresh}$ was calculated based on the maximum principal stress for traction separation when it is reached at a specific pore and Equation (11). The anisotropy of the microstructure was not considered since the simulation was a macroscale imitation of uniaxial fully sized fatigue specimens. $c_1$ and $c_2$ are the materials constants for fatigue crack initiation which have been assigned to zeros in the crack propagation analysis.

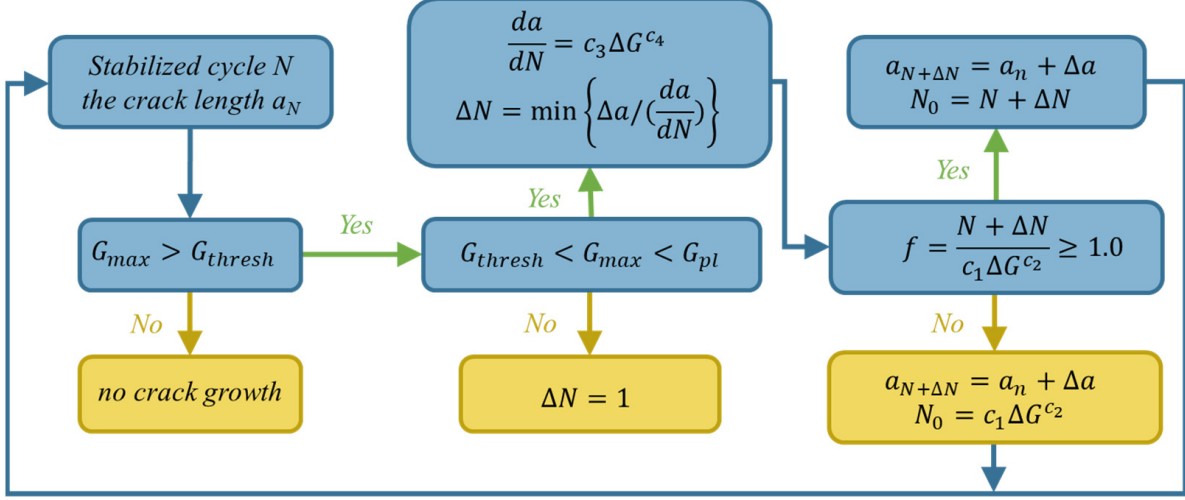

**Figure 7.** Schematic flow of the computation scheme.

## 3. Results

The tensile strength of AlSi10Mg was 451.1 MPa, which is greater than the cast counterpart [21], while the tensile strength of Ti-6Al-4V with the employed parameters was 1280.6 MPa, which is more than the wire + arc additively manufactured counterpart [11].

Strain localization at melt tracks leads to nuclei that propagate across the interface in mode I fracture, which lacks ductility. Please see [27] for further information on the specimens' quasi-static and cyclic deformation behavior. The damage evolution curves for fatigue loading at 20 and 5 Hz are shown in Figure 8. Using a Python script, we extracted the fracture area and cycle number for each frame of the simulation after the simulation. Both alloys exhibited stable damage evolution at the beginning with a low damage evolution rate. Afterward, AlSi10Mg started to experience continuous incremental changes in damage evolution rates. On the other hand, beyond 70% of a specimen's life, the damage evolution transferred into high-rate damage in Ti-6Al-4V. Due to geometric, load, and mechanical property differences, the alloys presented different transition mechanisms from short to long crack propagations rates. At a frequency of 20 kHz, the loading amplitude was shown in Figure 9 to have an effect on crack development. In the beginning, the fractures spread very slowly, but when $da/dN$ grew, the growth pace increased as well. Since $G_{pl}$ was beyond its upper limit, it can be seen that the slope is exceptionally steep, and the curve starts to follow the fatigue fracture development curve in area III, rather than the Paris law in region II. For example, mesh refinement is not required to run XFEM. By computing the strain values to determine the weakest point, the corresponding strain damage start criteria effectively produced the beginning crack. Because the pore had the lowest cross-sectional area, the first fracture was created there. There was no growth since the maximal energy release rate was so much lower than the critical energy release rate. It is clear that AlSi10Mg had three separate damage stages compared to the two phases of Ti-6Al-4V. Figure 9 show that the expansion of the fracture area in Ti-6Al-4V specimens was modest and steady over more than 90% of the specimen's life. The numerical results obtained here correspond well with experimental observations by the authors, see Table 3, which were published in [11,21,27,28].

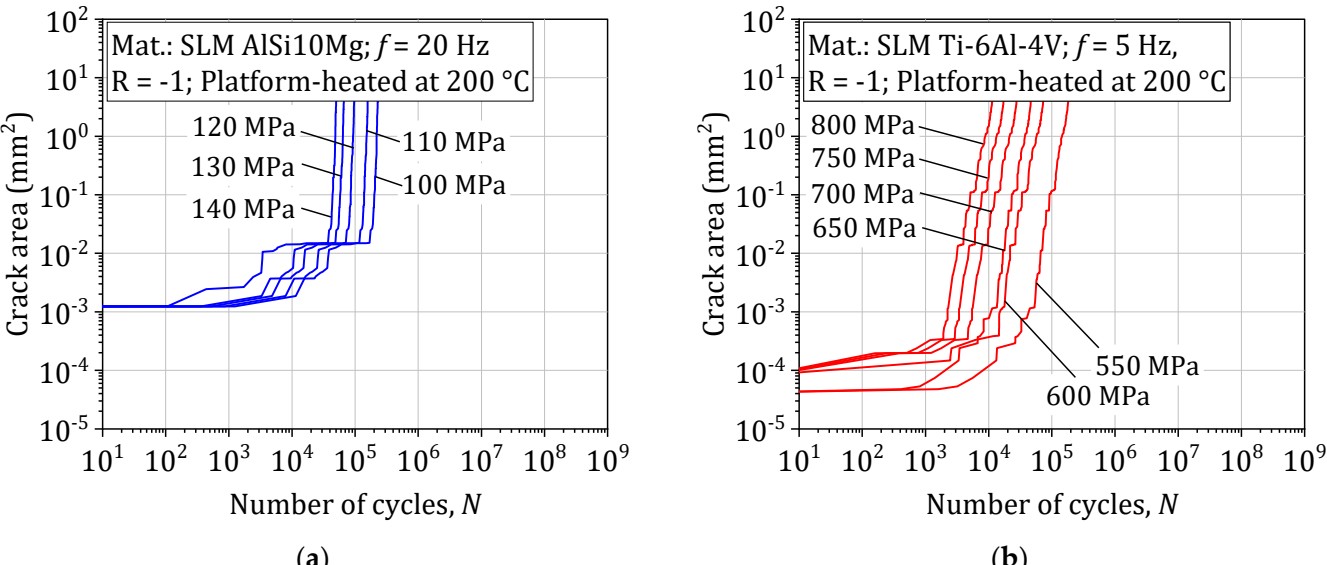

**Figure 8.** Influence of loading amplitude on the crack area evolution during low-frequency fatigue loading: (**a**) AlSi10Mg and (**b**) Ti-6Al-4V.

In Figures 10 and 11, the crack propagation curves are represented as a relationship between the development rate in the cross-sectional damaged area and the macroscopic stress intensity factor. The stress intensity factor is calculated according to the relation

$$\Delta K = \frac{2}{\pi} \sigma \sqrt{\pi \sqrt{area}} \qquad (17)$$

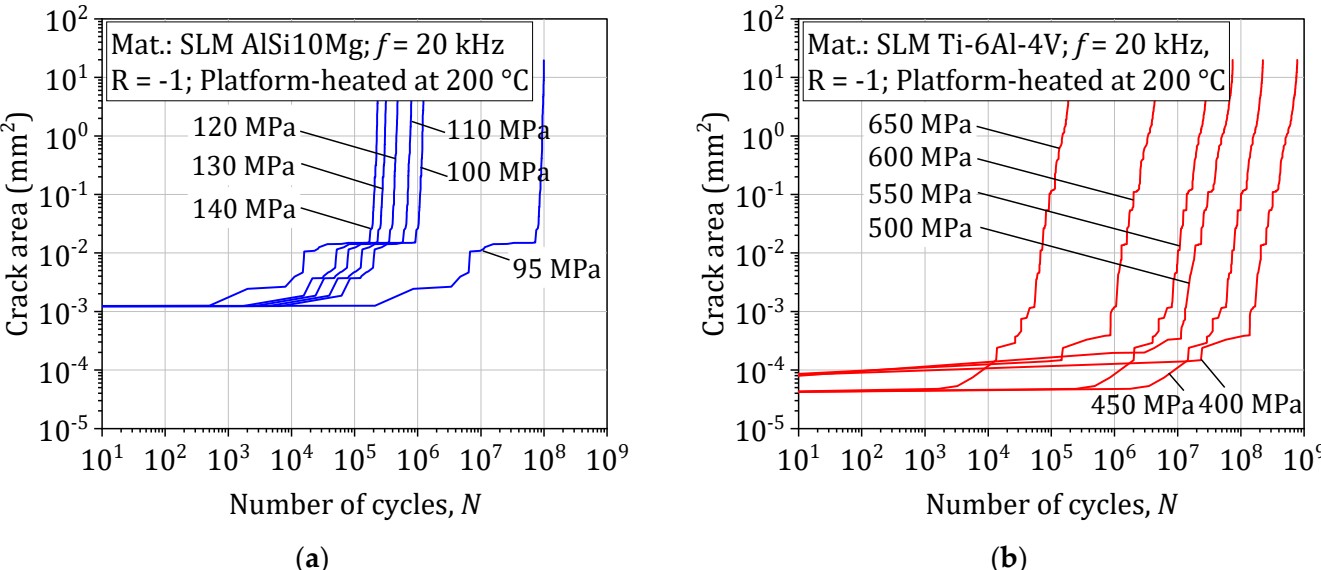

**Figure 9.** At 20 kHz, the loading amplitude has a significant impact on the propagation of cracks: (**a**) AlSi10Mg and (**b**) Ti-6Al-4V are available.

**Table 3.** Comparison between experimental and numerical fatigue data.

|  | **Experimental** | **Numerical** |
|---|---|---|
| **AlSi10Mg** | 41,288 | 51,463 |
| 140 MPa |  |  |
| 100 MPa | 220,475 | 234,895 |
| **Ti-6Al-4V** |  |  |
| 800 MPa | 13,113 | 12,316 |
| 500 MPa | 51,159,835 | 76,693,033 |

where $\Delta K$ is the stress intensity factor in mode I and $\sigma$ is the effective stress. It is based on the stress intensity factor formulations for a penny-shaped crack which can be found in [29]. It could be observed in the case of AlSi10Mg that $\Delta K$ is unstable in the region of short to long crack transition. Moreover, the propagation rates transfer to lower rates without pronounced effects on the threshold stress intensity factors. However, in Ti-6Al-4V, the values of $\Delta K$ are stable in the transition regions. Moreover, the threshold values shift to lower values in the ultrasonic frequency simulation compared to the low-frequency simulation, in addition to the lower rates. The findings also imply that the crack propagation curves of porous materials are less unique than bulk materials. The influence of frequency appears here to be most significant, and it is related to the energy which the material has to exert in every cycle that is much lower at higher frequencies. The observation is consistent with the literature [36]. The crack propagation rate decreases since one cycle consumes much less energy of the potential energy of the material.

The rapid decrease in crack propagation rate at the beginning as $\Delta K$ increases to a minimum, then the rate builds up again due to the transition from a short crack to a long crack propagation mechanism. This phenomenon can be seen qualitatively at low frequency in Figure 10 and at high frequency in Figure 11 with two different quantitative amounts.

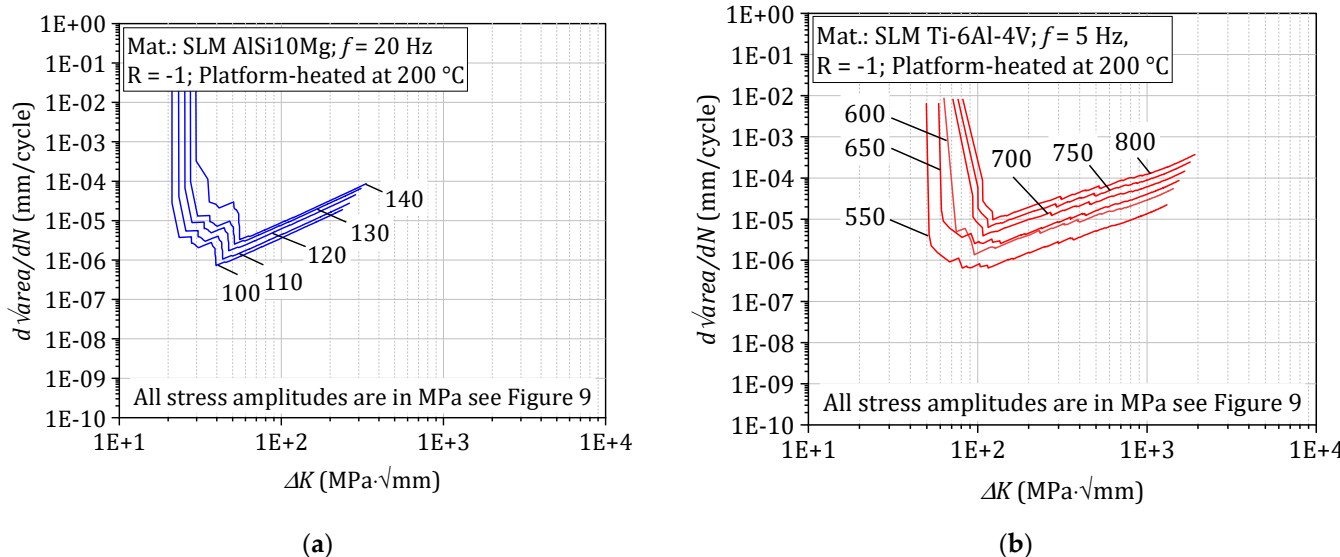

**Figure 10.** Crack propagation rates at: (**a**) AlSi10Mg 20 Hz and (**b**) Ti-6Al-4V 5 Hz.

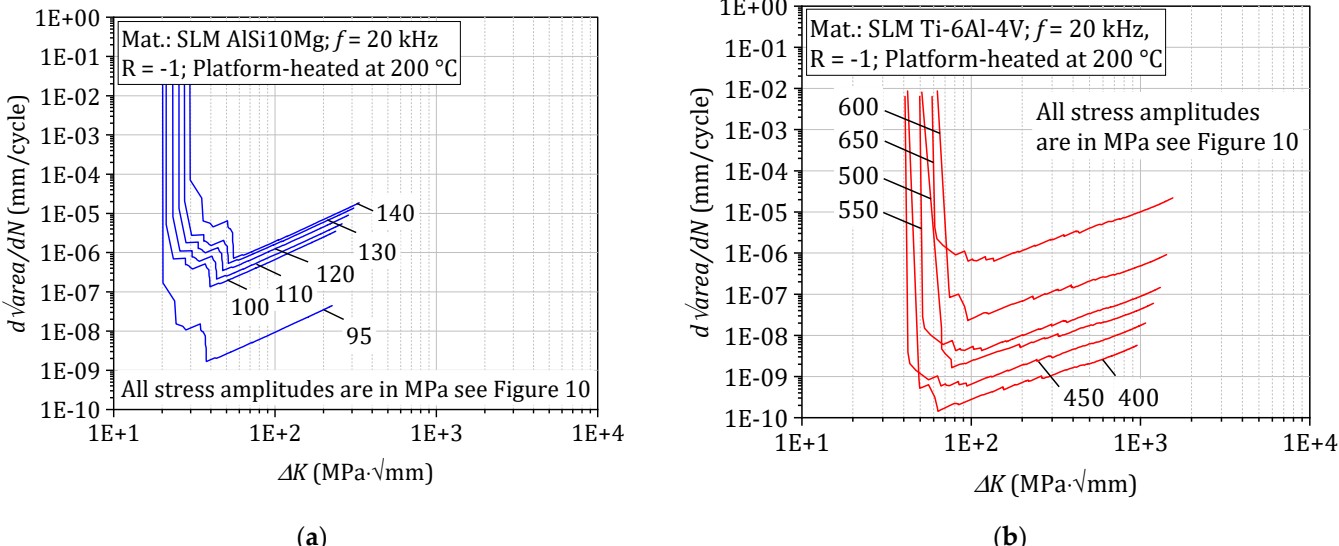

**Figure 11.** Crack propagation rates at 20 kHz: (**a**) AlSi10Mg and (**b**) Ti-6Al-4V.

## 4. Discussion

In ref [37], the implementation of the finite element method (FEM) and the extended finite element method (XFEM) to model fatigue crack propagation was implemented, similar to the concept applied here. It is claimed that FEM mesh only complies with the fatigue crack, while in XFEM, the crack is completely modeled, independent of the mesh. This was a great advantage to avoid mesh complications and inaccuracies in the calculated results. It was also concluded that the domain integral method is often used in FEM and XFEM and is preferred for evaluating the stress intensity factor. However, the displacement method is still often chosen to be used in practice. Since the stress intensity factor was not calculated in the simulation itself but in post-processing, XFEM achieved the study objectives to a great extent. It was mentioned that the maximum tangential stress criterion is used for determining the crack propagation path in most of the analyses [37]. However, the maximum principal strain was applied in the current study since the simulation was stress controlled. Additionally, contrary to FEM, local re-meshing must be used to advance the crack under linear elastic fracture mechanics (LEFM) conditions, which was avoided here by the application of XFEM. At the same time, either the cohesive elements or node

release technique are used under the conditions of elastic-plastic fracture mechanics. It was also stated that in XFEM, the level set method is used to describe the crack geometry, which was similarly applied in this investigation to deduce crack area evolution. In ref [38], the Symmetric Galerkin Boundary Element Method (SGBEM), and the SGBEM-FEM alternating/coupling methods, are compared with the common Extended Finite Element Method (XFEM) for modeling fracture and fatigue crack propagation in advanced structural components. It is claimed that the SGBEM-based methods are much more accurate than XFEM in terms of calculating the stress intensity factors and the fatigue–crack–growth rates as well. This shortcoming is circumvented by reliance on post-processing techniques as well as energy release rate as crack advancement criterion, especially under fatigue conditions. It is also shown that it requires coarser meshes than in XFEM. The author also claims that the methods mentioned in his paper outperform the XFEM for analyzing fracture and non-planar fatigue crack propagation in complex structural components. He also published a supplementary paper [39] that demonstrated the implementation of the methods but for 3D structural components. However, no comparison is made in [38,39] to experimental results.

In ref [40], the numerical simulation and validation of a fatigue crack propagation test are discussed. The test is carried out for a crack of a semi-elliptical shape located on a beam of a rectangular section subjected to four-point bending. An Instron 8874 biaxial testing machine was used to carry out the experimental tests, which were used for the purpose of validation, while the numerical simulation used the XFEM-based LEFM approach implemented in Abaqus [34]. It was found that the capabilities of the XFEM-based LEFM approach to simulate fatigue crack growth are validated, especially in complex crack fronts. This was similarly confirmed in the current investigation. It is also claimed that the geometry of the crack and the number of fatigue life cycles obtained in the numerical simulations were nearly identical to the actual test results. This agrees well with the findings here, which are compared to the results of [27]. The authors of [41] proposed a methodology for simulating the thermal fatigue delamination in composites. Additionally, it can predict both the delaminated crack opening displacement (DCOD) and the composite laminate permeability. A key advantage of this method is that it does not require an a priori definition of initial crack length or the crack propagation path. Moreover, it is claimed that both inter- and intralaminar crack growth in two- and three-dimensional geometries can be simulated.

The study of [42] proposed a modification for the existing XFEM methods. The study presented a so-called combined approximations (CA) approach, which is combined with XFEM for simulating the fatigue crack growth. The accuracy and efficiency of the proposed algorithm were examined. Moreover, some remarks were highlighted. For instance, the basis vectors can be easily computed at a very low computational cost. These findings represent a very suitable outlook for the current study. Additionally, the algorithm can still produce accurate results using only a small number of basis vectors. In the study [43], the extended finite element method (XFEM) capabilities available in Abaqus [34] are utilized for the computation of the stress intensity factor (SIF) in metallic lugs, which are made of Aluminum 7075-T6. Experimental tests were conducted for two load ratios, 0.1 and 0.5, respectively. It was found that the usage of different loading boundary conditions could highly affect the estimated fatigue life.

The study of [44] proposed a method to analyze the inter-layer defects inside slab tracks. It was concluded that the discontinuity problems in analyzing the inter-layer defects could be avoided when using XFEM. XFEM analysis also showed that at the initial stages, the gap area rate (GAR) value of the gap at the corner moderately increases but rapidly grows during the late stages, similar to the current study's findings with respect to crack area evolution under fatigue loading. While at the edges, the GAR increases significantly in the initial stages and increases linearly during later stages. In comparison, the authors of [45] introduced a study that discusses the limitations and capabilities of existing fatigue analysis standards. For instance, the direct cyclic method with XFEM in Abaqus was used

for the simulation of low cycle fatigue for compact-tension (CT) specimens and pipelines under the assumption of linear elastic material behavior. It was concluded that there is a huge potential for the direct cyclic method for fatigue analysis of elastic–plastic materials, as it shows significant agreement with the analytical solution in the high cycle fatigue (HCF). This was also shown to be true via the application of this same exact method for simulations of ultrasonic tests in the very high cycle fatigue regime (VHCF). While according to the author, the built-in direct cyclic method capabilities are limited to LCF analysis. The contrary was shown in the current study. In the investigation of [46], the results of a group of high cycle fatigue (HCF) experiments on steel joints were discussed. The tests were carried out under constant amplitude three-point bending, and the extended finite element method (XFEM) was later implemented for the simulation of the experimental tests. The authors illustrated that the XFEM predictions were successfully comparable to those obtained in the experimental tests. This complies very well with what the authors present for additively manufactured materials. For instance, the mean errors in the XFEM predictions for the fatigue life lay between the range of -20.7% to +0.9%. Moreover, the crack growth rate, number of cycles, crack shape, and final crack size comply with the experimental data [46].

In the study [47], a new predictive technique for a railway axle is discussed. It studies the application of fatigue crack propagation, which is subjected to fatigue bending load. Moreover, the authors demonstrated that this study is pioneering the numerical study of predicting fatigue crack trajectory, which appears in the model of a wheel of a railway axle. It was concluded that the numerically predicted, using Abaqus, and the experimentally measured crack trajectory in the axle is very comparable. This validation is made in the current study by comparing the outcome of XFEM simulations with the experimental results from [27]. Meanwhile, the study of [48] applied XFEM combined with a new cyclic cohesive zone model (CCZM). The approach tackles the simulation of fatigue crack propagation under mixed-mode loading conditions. It was concluded that the results of the performed uniaxial fatigue tests, which were used to build both the S-N curve and Goodman diagram, coincide with the experimental results. Additionally, the new damage accumulation equation is claimed to be more reasonable for finite cycle fatigue computation. It was also shown that the experimentally obtained Paris' law could be qualitatively reproduced using the cyclic cohesive zone model. This was similarly implemented here through the deduction of $d\sqrt{area}/dN$ curves for mode I stress intensity factor. However, more experimental and numerical investigations are required for quantitative comparison. Therefore, the authors plan in the outlook to apply the potential drop method for validation of crack length. In ref [49], XFEM was used for the purpose of determination of stress intensity factors (SIFs) in the wing-fuselage attachment lug, which can be found in light aerobatic aircraft. It was found that the predicted number of cycles to failure was relatively low, which was expected.

In the study [50], the simulation of fatigue crack growth in functionally graded materials (FGM) and plastically graded materials (PGM) is presented using the J-integral decomposition method and extended finite element method (XFEM). The fatigue crack growth rate is computed based on the Paris law's stress intensity factor, in contrast to the use of energy release rate in the current study. In contrast, the stress intensity factor is estimated by the J-integral decomposition method; for instance, the issues of the evaluation of stress at spatial mirror point and the strain energy density derivative. It was concluded that the crack deviates toward less stiff material in case the crack is not in a sequential direction. This was experimentally found using post-mortem μ-CT by Awd et al. [21]. The geometry of the fatigue crack is shown in Figure 12, where the crack initiates from a subsurface pore (~40 μm) and propagates to the surface. Afterward, a crack propagates towards the insides of the specimen. The experimental crack geometry can be seen in Figure 13. In [51], the crack propagation was numerically studied for a cruciform specimen. The study took place under out-of-phase biaxial fatigue loads. Moreover, the influence of phase angle of loading and displacement ratio has been presented as well. XFEM was used for this purpose and by

applying different criteria from the literature for non-proportional loading. The XFEM has facilitated a parametric study to be carried out. In addition, symmetrical branching was predicted for an initial crack inclined at the phase angle of loading at 45°, 90°, and 180°. It was found that the numerical results are comparable to the experimental observations found in the literature. However, the study reveals important differences depending on the different orientation criteria in the crack path predictions. It is worth mentioning that during this discussion that the XFEM methods were not yet used to study the influence of loading frequency on crack propagation rate, hence the absence of frequency from the discussion on XFEM in the literature.

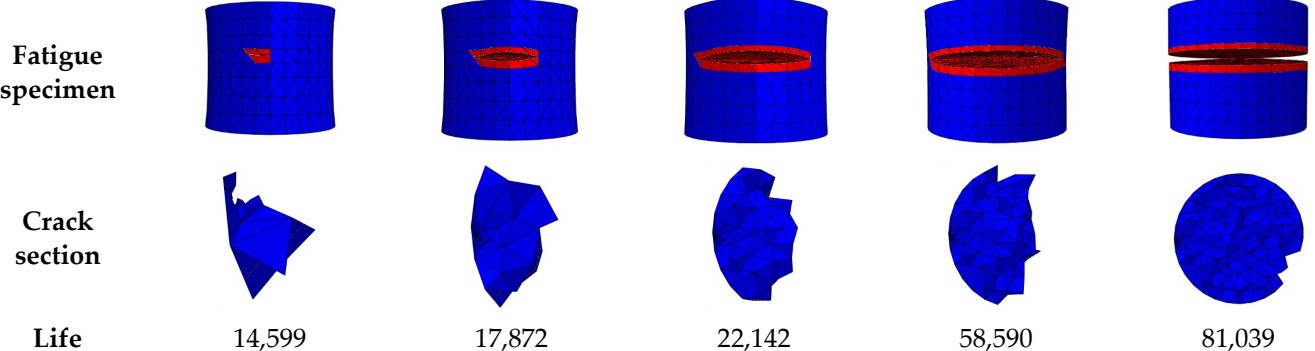

| | | | | | |
|---|---|---|---|---|---|
| **Fatigue specimen** | | | | | |
| **Crack section** | | | | | |
| **Life** | 14,599 | 17,872 | 22,142 | 58,590 | 81,039 |

**Figure 12.** The geometry of the crack during its fatigue propagation as obtained numerically of Ti-6Al-4V at 800 MPa of 5 Hz.

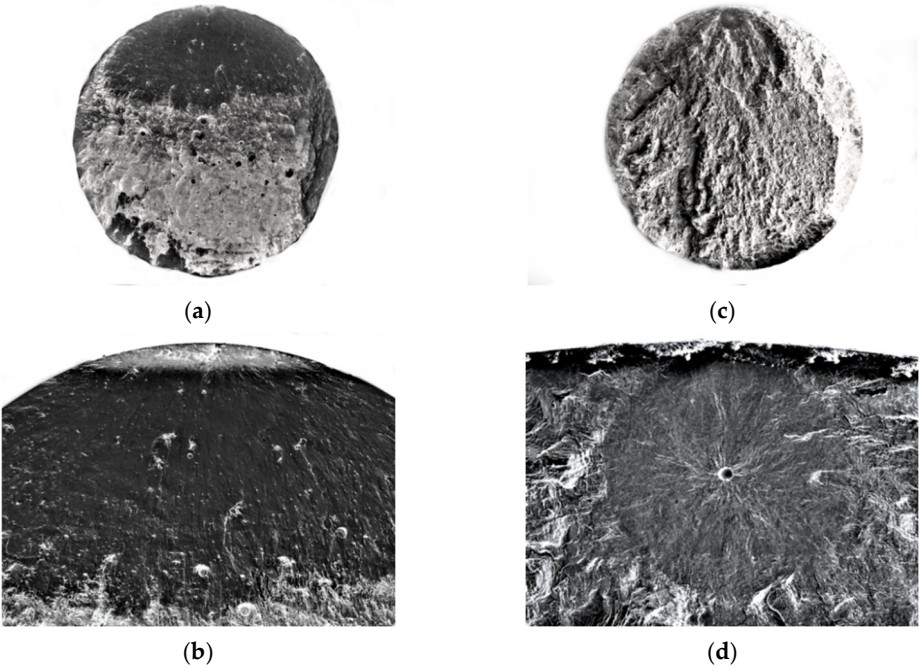

**Figure 13.** Fractographich morphology of fatigue cracking showing crack geometry: (**a**,**b**) AlSi10Mg —120 MPa; (**c**,**d**) Ti-6Al-4V—800 MPa.

## 5. Summary

Cylindrical specimens for AlSi10Mg and Ti-6Al-4V were developed utilizing modified SLM 250 HL and 500 HL equipment. Elastoplastic properties of metals and alloys were extracted from the obtained data using an instrumented indentation method. Damage evolution curves for fatigue loading at 20, 5 Hz, and 20 kHz were deduced based on XFEM.

AlSi10Mg has three separate damage stages compared to the two phases of Ti-6Al-4V. Both alloys exhibit stable damage evolution at the beginning with low damage evolution

rates. XFEM was validated by comparison with the experimental data for modeling fatigue crack propagation. Accuracy was enhanced here using post-processing techniques such as energy release rate.

Numerical simulation and validation of a fatigue crack propagation test were presented and discussed in view of actual literature in a contemporary manner. It was found that the capabilities of the XFEM-based LEFM approach to simulate fatigue crack growth were validated. The geometry of the crack and the number of fatigue life cycles obtained in the numerical simulations were nearly identical to the actual test results. The proposed method of analysis in additive manufacture for the inter-layer defects inside melt tracks was also well qualified.

There is huge potential for the indirect cyclic method for fatigue analysis of elastic-plastic materials. XFEM combined with classic fracture mechanical post-processing tackled the simulation of fatigue crack propagation under mixed-mode loading conditions. The simulation of fatigue crack growth is based on the energy release rate method and extended finite element method. It was found that the predicted number of cycles to failure was relatively accurate, which will be used to further calibrate fatigue lifetime prediction models.

**Author Contributions:** Conceptualization, M.A.; methodology, M.A.; software, M.A.; validation, M.A.; formal analysis, M.A.; investigation M.A.; resources, F.W.; data curation, M.A.; writing—original draft preparation, M.A.; writing—review and editing, F.W.; visualization, M.A.; supervision, F.W.; project administration, F.W.; funding acquisition, F.W. All authors have read and agreed to the published version of the manuscript.

**Funding:** The authors thank the German Research Foundation (Deutsche Forschungsgemeinschaft, DFG) for its financial support within the research project "Mechanism-based understanding of functional grading focused on fatigue behavior of additively processed Ti-6Al-4V and Al-12Si alloys" (WA 1672/25-1).

**Institutional Review Board Statement:** Not applicable.

**Informed Consent Statement:** Not applicable.

**Data Availability Statement:** The meta-data is protected under non-disclosure agreements.

**Acknowledgments:** The authors further thank Fraunhofer IAPT, Hamburg, for the provision of the samples in the framework of excellent scientific collaboration.

**Conflicts of Interest:** The authors declare no conflict of interest.

**Sample Availability:** Samples of the compounds are available from the authors.

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
