# Peer review of "Numerical Investigation of the Influence of Fatigue Testing Frequency on the Fracture and Crack Propagation Rate of Additive-Manufactured AlSi10Mg and Ti-6Al-4V Alloys"

_solids, doi:10.3390/solids3030030_

Round 1
Reviewer 1 Report
This article presents a study of fatigue crack growth in two additively manufactured alloys: AlSi10Mg alloy and Ti-6Al-4V. The study is based on prior work by the authors. Material properties are determined by instrumented indentation. The sample geometry and porosity are determined by microcomputed tomography and crack growth is simulated with XFEM, based on Paris Law. The study focusses on the effect of loading amplitude and frequency on crack growth. The combination of experimental and theoretical components of the work is of interest and the article is publishable once the following issues are addressed:
1. The article lacks clarity, with few details provided related to all components other than the technique used to advance the crack. It is suggested that Section 2 may be divided in subsections, each addressing material properties identifications, geometry (and microstructure) identification and the methods used for modeling crack growth, with adequate details presented in each section. This will greatly benefit the article.
2. Material properties are identified using instrumented indentation. While the identification of the Young Modulus using Oliver -Pharr is standard, the identification of viscoplastic properties is not as straightforward. In fact, the literature discusses that the inverse problem for the plastic properties is ill posed with multiple solutions being possible. The article does not detail how the solution is stabilized and how the inferred properties for the additively made alloys of study differ from the properties of their cast versions. Including a properties table is highly recommended. Likewise, describing how the strain rate sensitivity is captured, is also recommended, considering the main objective of the study.
3. Microcomputed tomography has been used to identify pores and shape of the surface. The information provided regarding this aspect is sketchy. Please clarify whether any information about the microstructure beyond porosity is captured.
4. Please provide parameters c3 and c4 used in Eq 11, Paris Law.
5. Figure 7 and the paragraph above. Please provide a more detailed description of the flow diagram, particularly the right-side column. Parameter f in Figure 7 seems to be distinct from parameter f in Eq 15, please, check. Please, specify the meaning of parameters c1 and c2 in the right column of Figure 7.
6. G_thresh and G_pl are parameters used to determined the applicability of Paris law. How are these determined from experiment? How was the anisotropy (or texture) of the AM specimens taken into account?
7. Figure 11. Why the curves increase rapidly as DK decreases? Shouldn’t they decrease at DK threshold? Please explain the meaning.
Thank you
Reviewer 2 Report
-

Round 2
Reviewer 2 Report
-

Author Response
Reviewer (2) – Yello Highlights in the revised manuscript 2
Comment: The pore fraction and the specimen diameter must be indicated in the manuscript.
Answer: See lines 85-86.
Comment: In table 2, the units of c3 must be indicated in the manuscript.
Answer: See table 2.
Comment: It should be indicated in the manuscript how these parameters (c1, c2, c3. c4, Gthresh,…) have been obtained from the instrumented indentation tests.
Answer: See lines 220-233.
Comment: Under linear elastic fracture mechanics (LEFM) conditions, why, if ?max> ??l does fracture not occur?
Answer: See lines 224-226.
Comment: According to the results, do the porous materials not have a da/dN-ΔK curve characteristic of the material? The cause of this should be discussed in the manuscript.
Answer: See lines 280-285.
Comment: What causes in modeling the change in fatigue behavior of this material with
frequency? What are material properties key to this case?
Answer: See table 2.
Comment: Since the finite element method does not have specific units (it is enough that there is coherence between them), the effect of frequency on the results is not understood if
there is no material parameter that depends on time. What are these parameters?
Answer: See table 2.
Comment: The discussion section does not address the paper's aim, which is the influence of fatigue test frequency on the fracture rate and cracks propagation of AlSi10Mg and Ti-6Al-4V additive-fabricated materials. It is not understood that the discussion section addresses the technique used. The experimental results of the fatigue
Answer: The paper aims to prove the possibility of using the technique to answer the question of the influence of frequency. The influence of frequency is not introduced here. It is well established in the literature, and the introduction shows that. The article focuses on a technique to show the influence of frequency. Thus, the trustworthiness of the technique is the point of discussion, not the influence of frequency which is not debatable.
Comment: The experimental results of the fatigue tests (crack geometry and several fatigue life cycles) should appear in the manuscript to validate the numerical results.
Answer: See table 3 and figure 13.
Comment: The manuscript should add a comparison of numerical and experimental crack geometry and the number of fatigue life cycles.
Answer: See table 3.
Comment: The geometry of the crack front (numerical) is very irregular. Furthermore, according to Fig. 12, the cracks appear superficial (not interior). The expression used to calculate the stress intensity factor does not seem adequate.
Answer: See lines 411-414; figures 12 and 13.